# Cryo-EM structure of the endothelin-1-ET$_B$-G$_i$ complex

**Fumiya K Sano, Hiroaki Akasaka, Wataru Shihoya*, Osamu Nureki***

Department of Biological Sciences, Graduate School of Science, The University of Tokyo, Tokyo, Japan

**Abstract** The endothelin ET$_B$ receptor is a promiscuous G-protein coupled receptor that is activated by vasoactive peptide endothelins. ET$_B$ signaling induces reactive astrocytes in the brain and vasorelaxation in vascular smooth muscle. Consequently, ET$_B$ agonists are expected to be drugs for neuroprotection and improved anti-tumor drug delivery. Here, we report the cryo-electron microscopy structure of the endothelin-1-ET$_B$-G$_i$ complex at 2.8 Å resolution, with complex assembly stabilized by a newly established method. Comparisons with the inactive ET$_B$ receptor structures revealed how endothelin-1 activates the ET$_B$ receptor. The NPxxY motif, essential for G-protein activation, is not conserved in ET$_B$, resulting in a unique structural change upon G-protein activation. Compared with other GPCR-G-protein complexes, ET$_B$ binds G$_i$ in the shallowest position, further expanding the diversity of G-protein binding modes. This structural information will facilitate the elucidation of G-protein activation and the rational design of ET$_B$ agonists.

## Editor's evaluation

By combining NanoBiT tethering and FSEC methodology, the authors present a useful strategy for the efficient expression and purification of the GPCR-G-protein complex, as demonstrated here by the cryo-EM structure of human ETB in complex with the vasoconstrictioning peptide ET-1 and the inhibitory G-protein (Gi). The ETB-Gi-protein complex provides valuable and convincing data on how ET-1 binding is coupled to Gi-protein binding. The complex structure is solid and will appeal to the GPCR and pharmacology communities.

*For correspondence:
wtrshh9@gmail.com (WS);
nureki@bs.s.u-tokyo.ac.jp (ON)

## Introduction

Endothelins are 21-amino-acid vasoconstricting peptides produced primarily in the endothelium *Yanagisawa et al., 1988* and play a key role in vascular homeostasis. Among the three endothelin isopeptides (ET-1–3), ET-1 was first discovered as a potent vasoconstrictor *Maguire and Davenport, 2014*; *Davenport et al., 2016*. ET-1 transmits signals through two receptor subtypes, the ET$_A$ and ET$_B$ receptors, which belong to the class A G-protein-coupled receptors (GPCRs) *Arai et al., 1990*; *Sakurai et al., 1990*. The endothelin receptors exert regulatory control over cellular processes important for growth, survival, invasion, and angiogenesis *Haryono et al., 2022*; *Barton and Yanagisawa, 2019*. In the vascular system, the ET$_A$ receptor serves as the primary mediator of vasoconstriction, and its irreversible binding to ET-1 leads to prolonged vasoconstriction. By contrast, the ET$_B$ receptor primarily induces vasorelaxation via the nitric oxide-mediated pathway, and it acts as a clearance receptor that removes circulating ET-1 via the lysosomal pathway *Bremnes et al., 2000*. The ET$_B$ receptor is known to be a promiscuous GPCR, capable of activating multiple types of G-proteins *Inoue et al., 2019*; *Doi et al., 1999*. In vascular smooth muscle, it induces nitric oxide-mediated vasorelaxation via G$_q$ signaling. Furthermore, the astrocytic ET$_B$ receptor-mediated G$_i$ signal reduces intercellular communication through gap junctions *Tencé et al., 2012*, and the Rho signal in astrocytes leads to cytoskeletal

reorganization and cell-adhesion-dependent proliferation *Koyama and Baba, 1996*, promoting the induction of reactive astrocytes and neuroprotection *Koyama, 2021*.

Drug development targeting the endothelin receptors has primarily focused on antagonists owing to the vasodilation effect *Haryono et al., 2022*; *Barton and Yanagisawa, 2019*. Bosentan, the first non-peptide antagonist for $ET_A$ and $ET_B$, is currently in clinical use for the treatment of pulmonary arterial hypertension. Moreover, $ET_A$-selective antagonists are used as therapeutic agents with fewer side effects. Notably, endothelin-1 acts primarily through the $ET_A$ receptor and is implicated in the neoplastic growth of multiple tumor types. Thus, endothelin receptor antagonists such as atrasentan and zibotentan have demonstrated potential anticancer activity in preclinical studies *Rosanò et al., 2013*. The development of $ET_B$ agonists is also underway, as they provide therapeutic benefits such as vasodilation and neuroprotection *Davenport et al., 2016*; *Koyama, 2021*. IRL1620, a truncated analog of ET-1, is the smallest $ET_B$-selective agonist that has been shown to selectively and transiently increase tumor blood flow *Gulati et al., 2012*, making it a potential adjuvant cancer therapy for enhancing the delivery of anti-tumor drugs and acute ischemic stroke *Ranjan and Gulati, 2022*. However, IRL1620 is a linear peptide with exposed N- and C-termini and thus has problems in terms of pharmacokinetics and drug delivery. Currently, small-molecule $ET_B$-selective agonists have not been developed, hindering drug development targeting the $ET_B$ receptor.

To date, eight crystal structures of the human $ET_B$ receptor have been reported, elucidating the structure-activity relationships of the peptide agonists and small-molecule clinical antagonists *Shihoya et al., 2016*; *Nagiri et al., 2019*; *Izume et al., 2020*; *Shihoya et al., 2018*; *Shihoya et al., 2017*. Nevertheless, the detailed activation mechanism has remained elusive due to the crystallization constructs containing T4 lysozyme in the intracellular loop (ICL) 3 and thermostabilizing mutations *Okuta et al., 2016*; *Nakai et al., 2022* that stabilize the inactive state. Moreover, the conserved $N^{7.49}P^{7.50}xxY^{7.53}$ motif (superscripts indicate Ballesteros–Weinstein numbers *Ballesteros and Weinstein, 1995*) essential for G-protein activation *Venkatakrishnan et al., 2013* is altered to $N^{7.49}P^{7.50}xxL^{7.53}Y^{7.54}$ in the wild-type $ET_B$ receptor. Thus, little is known about $ET_B$-mediated G-protein activation. Here, we report the 2.8 Å-resolution cryo-electron microscopy (cryo-EM) structure of the human $ET_B$-$G_i$ signaling complex bound to ET-1, revealing the unique mechanisms of receptor activation and G-protein coupling.

## Results
### Development of fusion-G system for structural determination

For the cryo-EM analysis, we initially used the thermostabilized receptor $ET_B$-Y5, which contains five thermostabilizing mutations *Okuta et al., 2016*. However, the purified $ET_B$-Y5 could not form a stable complex with the $G_i$ trimer because the mutations were known to stabilize the inactive conformation. Thus, we chose to use the wild-type $ET_B$ for the structural study. To purify the stable GPCR-G-protein complex, we developed a 'Fusion-G system' (*Figure 1A*) by combining two complex stabilization techniques. One of these techniques was the NanoBiT tethering strategy *Duan et al., 2020*; *Dixon et al., 2016*, where the large part of NanoBiT (LgBiT) was fused to the C-terminus of the receptor, and a modified 13-amino acid peptide of NanoBiT (HiBiT) was fused to the C-terminus of Gβ via the GS linker. HiBit has a potent affinity for LgBiT ($Kd$ = 700 pM) and thus provides an additional linkage to stabilize the interface between H8 of the receptor and the Gβ subunit of the G-protein. This strategy has been successfully used to solve several GPCR/G-protein complex structures *Duan et al., 2020*; *Xia et al., 2021*. The other technique was a 3-in-1 vector for G-protein expression, in which the Gα subunit was fused to the C-terminus of the Gγ subunit (*Kim et al., 2020*; *Nureki et al., 2022*; *Figure 1A*). The resulting pFastBac-Dual vector could produce a virus that expressed the G-protein trimer. Moreover, the protease-cleavable green fluorescent protein (EGFP) was connected to the C-terminus of the receptor-LgBiT fusion, allowing analysis of complex formation by fluorescence-detection size-exclusion chromatography (FSEC) *Hattori et al., 2012*. Using this system, we confirmed the complex formation of $LPA_1$ and $S1P_5$ with $G_i$ (*Figure 1—figure supplement 1A, B*), whose structures in complex with the $G_i$ trimer had previously been reported *Akasaka et al., 2022*; *Liu et al., 2022*; *Yuan et al., 2021*; *Xu et al., 2022a*.

We cloned the full-length human $ET_B$ receptor into the LgBiT vector. Using the fusion-G-system, we confirmed the formation of the $ET_B$-$G_i$ complex (*Figure 1B*). The co-expressed cells from a 300 ml culture were solubilized and purified by Flag affinity chromatography. After incubation with scFv16,

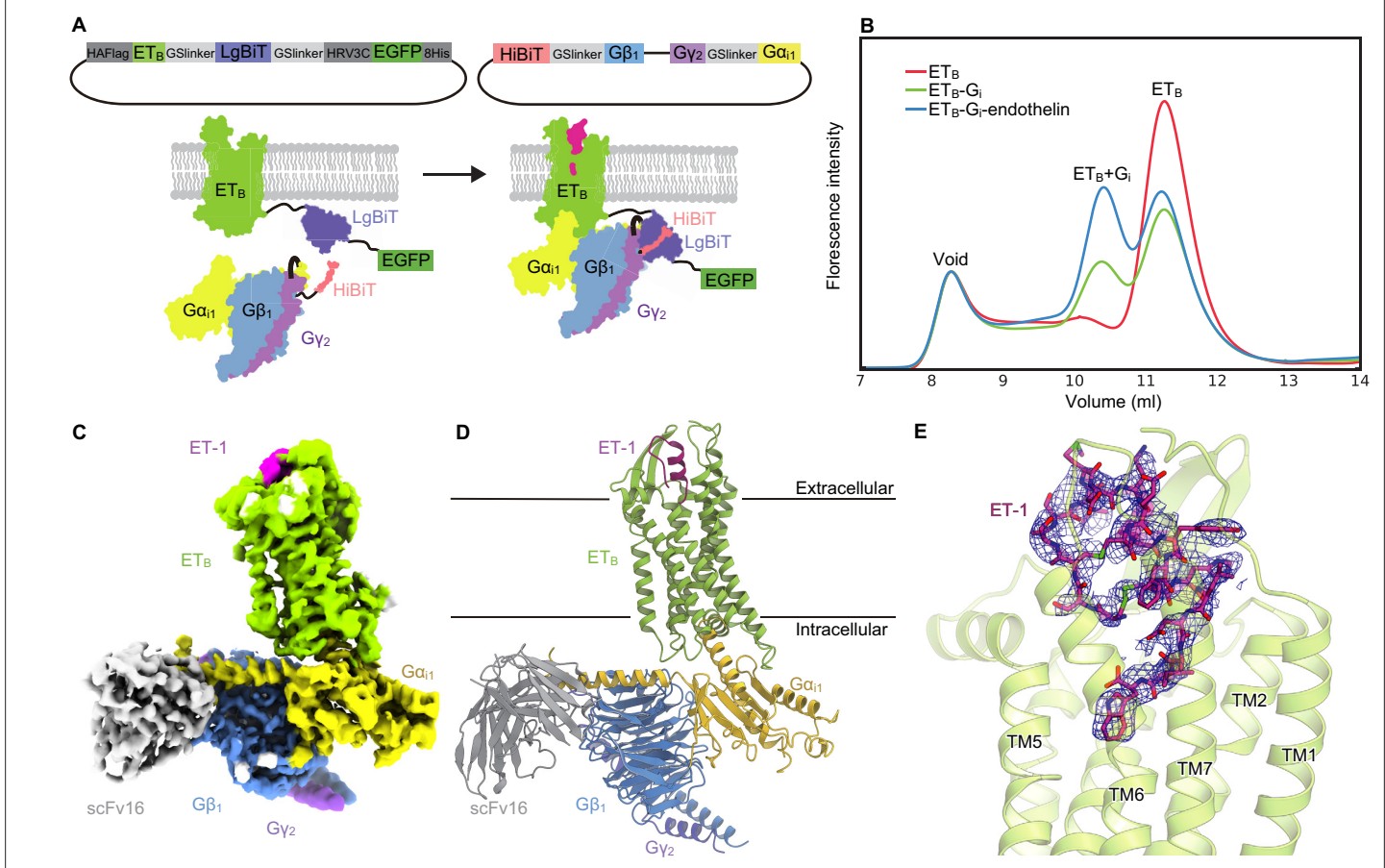

**Figure 1.** Overall structure of the ET-1-ETB-Gi signaling complex. (**A**) Schematic representations of the fusion-G system. (**B**) Fluorescence-detection size-exclusion chromatography (FSEC) analysis of complex formation by the $ET_B$ receptor. The fluorescence intensities are adjusted to equalize those corresponding to the void volumes. (**C**) Cryo-electron microscopy (cryo-EM) map with variously colored densities. (**D**) Structure of the complex determined after refinement in the cryo-EM map, shown as a ribbon representation. (**E**) Density focused on ET-1.

The online version of this article includes the following source data and figure supplement(s) for figure 1:

**Figure supplement 1.** Fusion-G system.

**Figure supplement 1—source data 1.** An uncropped image of a gel.

**Figure supplement 2.** Cryo-electron microscopy (Cryo-EM) workflow, maps, and model quality.

the complex was purified by size exclusion chromatography (*Figure 1—figure supplement 1C, D*). The structure of the purified complex was determined by single-particle cryo-EM analysis with an overall resolution of 2.8 Å (*Figure 1C and D*, *Figure 1—figure supplement 2*, *Table 1*). No density corresponding to NanoBiT was observed in the 2D class averages and reconstructed 3D density map, as in the previous structural studies using the NanoBiT tethering strategy *Duan et al., 2020*; *Xia et al., 2021*; *Wang et al., 2021*; *Xu et al., 2022b*. We refined with a mask on the receptor and obtained the receptor structure with a nominal resolution of 3.1 Å (*Figure 1—figure supplement 2*, *Table 1*). The agonist ET-1 is well resolved (*Figure 1E*).

## Receptor conformational changes upon $G_i$ activation

The extracellular half of the receptor superimposes well on the ET-1-bound crystal structure, and ET-1 interacts closely with the receptor in a similar manner (*Figure 2A*). Previous crystallographic analyses have suggested that the ET-1 binding leads to the downward movement of $N378^{7.45}$ and $W336^{6.48}$ in the $C^{6.47}W^{6.48}xP^{6.50}$ motif at the bottom of the ligand binding pocket, which ultimately results in the downward rotation of $F332^{6.44}$ in the $P^{5.50}I^{3.40}F^{6.44}$ motif, and leading to the intracellular opening *Shihoya et al., 2016*; *Nagiri et al., 2019*; *Izume et al., 2020*; *Shihoya et al., 2018*; *Shihoya*

**Table 1.** Cryo-EM data collection, refinement, and validation statistics.

| Data collection | $ET_B$-$G_i$ (overall) | $ET_B$-$G_i$ (receptor focused) |
|---|---|---|
| Microscope | Titan Krios (Thermo Fisher Scientific) | |
| Voltage (keV) | 300 | |
| Electron exposure (e⁻/Å2) | 49.965 | |
| Detector | Gatan K3 summit camera (Gatan) | |
| Magnification | ×105,000 | |
| Defocus range (μm) | −0.8–1.6 | |
| Pixel size (Å/pix) | 0.83 | |
| Number of movies | 10,408 | |
| Symmetry | C1 | |
| Picked particles | 3,863,134 | |
| Final particles | 260,085 | |
| Map resolution (Å) | 2.80 | 3.13 |
| FSC threshold | 0.143 | |
| **Model refinement** | | |
| Atoms | 9,367 | 2,523 |
| **R.m.s. deviations for ideal** | | |
| Bond lengths (Å) | 0.002 | 0.003 |
| Bond angles (°) | 0.56 | 0.52 |
| **Validation** | | |
| Clashscore | 11.68 | 7.7 |
| Rotamers (%) | 0.00 | 0.00 |
| **Ramachandran plot** | | |
| Favored (%) | 96.55 | 98.04 |
| Allowed (%) | 3.19 | 1.96 |
| Outlier (%) | 0.26 | 0.00 |

*et al., 2017*. The downward movement of the residues is larger in the $ET_B$-$G_i$ complex than in the ET-1-bound crystal structure (*Figure 2B*), which consequently results in the outward displacement of the intracellular portion of the transmembrane helix (TM) 6 by 6.8 Å as compared to the apo state (*Figure 2C*), and by 5.1 Å as compared to the ET-1-bound crystal structure (*Figure 2D*). The degree of TM6 opening observed is less than those of other $G_i$-coupled receptors (e.g. μOR: 10 Å, $CB_1$: 11.6 Å, and $S1P_1$: 9 Å) *Yuan et al., 2021*; *Zhuang et al., 2022*; *Hua et al., 2020*.

In most class A GPCRs, the highly conserved $D^{3.49}R^{3.50}Y^{3.51}$ and $N^{7.49}P^{7.50}xxY^{7.53}$ motifs on the intracellular side play an essential role in G-protein coupling *Venkatakrishnan et al., 2013*. The $D^{3.49}R^{3.50}Y^{3.51}$ motif is conserved in the $ET_B$ receptor. Upon receptor activation, the ionic lock between $D198^{3.49}$ and $R199^{3.50}$ is broken, and $R199^{3.50}$ becomes oriented towards the intracellular cavity (*Figure 2E*), similar to other GPCRs *Zhuang et al., 2022*; *Rasmussen et al., 2011*; *Ring et al., 2013*; *Cherezov et al., 2007*; *Manglik et al., 2012*; *Huang et al., 2015* (*Figure 2F and G*). In contrast, the $N^{7.49}P^{7.50}xxY^{7.53}$ motif is altered to $N^{7.49}P^{7.50}xxL^{7.53}Y^{7.54}$, where $Y^{7.53}$ is replaced by $L386^{7.53}$. In most class A GPCRs, receptor activation disrupts the stacking interaction between $Y^{7.53}$ and $F^{8.50}$ *Carpenter and Tate, 2017*. $Y^{7.53}$ moves inwardly and forms a water-mediated hydrogen bond with the highly conserved $Y^{5.58}$ (*Figure 2F and G*; *Ring et al., 2013*; *Huang et al., 2015*; *Deupi et al., 2012*). The mutations of the tyrosines significantly reduce G-protein activation *Goncalves et al., 2010*; *Flock et al., 2015*, indicating that the interaction between $Y^{5.58}$-and $Y^{7.53}$ stabilizes the active conformation of the receptor.

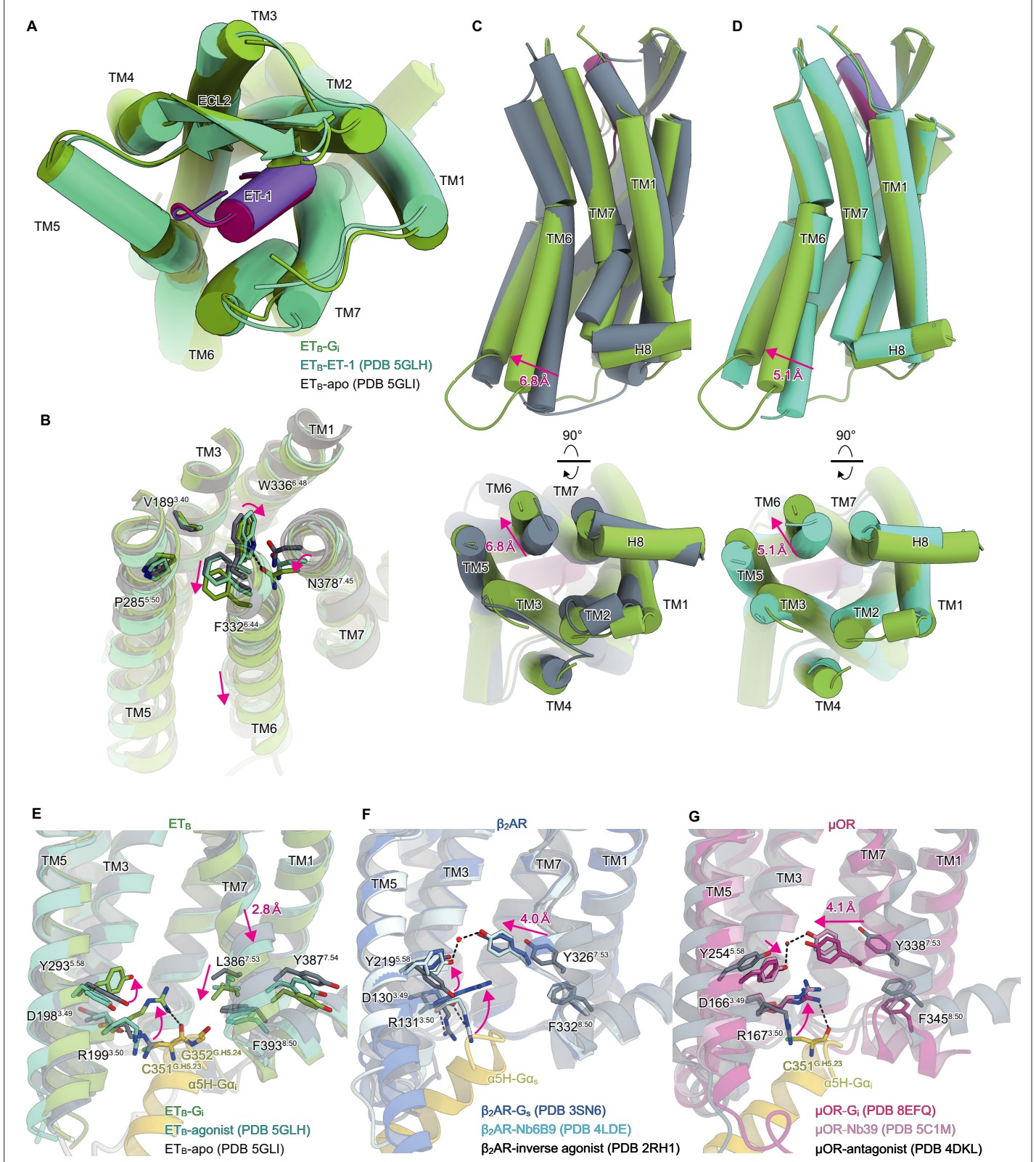

**Figure 2.** Structural changes upon G-protein activation. (**A**) Superimposition of the ET-1-bound receptor in the crystal and cryo-electron microscopy (cryo-EM) structures. (**B**) Superimposition of the ET$_B$ structures, focused on the receptor core. (**C, D**) Superimpositions of the G$_i$-complexed ET$_B$ structure with the ET-1-bound crystal structure (**C**) and apo structure (**D**). (**E-G**) D$^{3.49}$R$^{3.50}$Y$^{3.51}$ and N$^{7.49}$P$^{7.50}$xxY$^{7.53}$ motifs in ET$_B$ (**E**), β2AR (**F**), and μOR (**G**). Black dashed lines indicate hydrogen bonds.

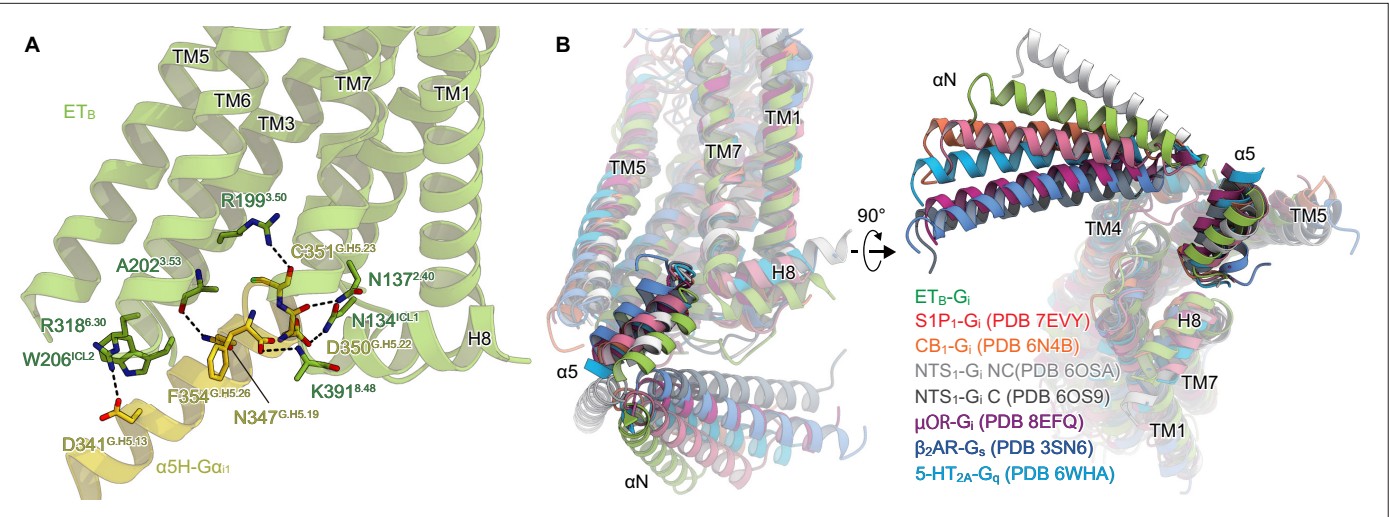

**Figure 3.** Comparison of the Gi binding modes. (**A**) Hydrogen-bonding interactions between ETB and the α5-helix, indicated by black dashed lines. (**B**) Comparison of the Gα positions in the GPCR-G-protein complexes. The structures are superimposed on the receptor structure of the NTS₁-C state.

The online version of this article includes the following figure supplement(s) for figure 3:

**Figure supplement 1.** Detailed ETB-Gi interface.

**Figure supplement 2.** Comparison of the interactions between the α5-helix and TM7-H8.

Along with the motion, the intracellular portion of TM7 is displaced by approximately 4 Å. As L386$^{7.53}$ is a hydrophobic residue, it cannot form a polar interaction, and hence in the ETB-Gi complex, TM7 is not displaced inwardly (***Figure 2C***). Nonetheless, the stacking interaction with F393$^{8.50}$ is disrupted similarly. Moreover, the intracellular portion of TM7 is displaced downward by 2.8 Å. As expected, Y$^{7.54}$ is directed towards the membrane plane, and its rotamer does not change upon receptor activation (***Figure 2E***). The substitution of Y$^{7.53}$ with L386$^{7.53}$ affects the movement of TM7 upon receptor activation, thereby distinguishing it from other GPCRs.

## Shallow Gi coupling

These conformational changes create an intracellular cavity for G-protein recognition (***Figure 3A***, ***Figure 3—figure supplement 1***). The cavity closely interacts with the C-terminal α5-helix of Gαi, the primary determinant for G-protein coupling. In particular, R199$^{3.50}$ forms a hydrogen bond with the backbone carbonyl of C351$^{G.H5.23}$ (superscript indicates the common Gα numbering [CGN] system ***Flock et al., 2015***), which is commonly observed in other GPCR-Gi complexes ***Yuan et al., 2021***; ***Zhuang et al., 2022***; ***Hua et al., 2020***; ***Kato et al., 2019***. Additionally, the C-terminal carboxylate of α5-helix forms electrostatic interactions with the backbone nitrogen atom of K391$^{8.48}$. There are several other hydrogen-bonding interactions between the α5-helix and the receptor (***Figure 3A***). On ICL2, W206$^{ICL2}$ fits into a hydrophobic pocket formed by L194$^{G.S3.01}$, F336$^{G.H5.08}$, T340$^{G.H5.12}$, I343$^{G.H5.15}$, and I344$^{G.H5.16}$ in the Gαi subunit. Moreover, ICL2 forms extensive van der Waals interactions with the αN (***Figure 3—figure supplement 1***), which are not observed in other GPCR-Gi complexes (***Figure 3—figure supplement 1***; ***Yuan et al., 2021***; ***Hua et al., 2020***).

Upon receptor activation, the intracellular portion of TM7 moves downwards, resulting in the unique Gi coupling mode. Along with this motion, L386$^{7.53}$ directly forms a hydrophobic contact with G352$^{G.H5.24}$. Moreover, TM7 and H8 extensively interact with the α5-helix (***Figure 3—figure supplement 2***), which is not observed in other GPCR-Gi complexes (***Figure 3—figure supplement 2***; ***Yuan et al., 2021***; ***Zhuang et al., 2022***; ***Hua et al., 2020***; ***Kato et al., 2019***). These structural features enable the α5-helix of ETB-Gi to be located in the shallowest position relative to the receptor among the Gs, Gi, and Gq-coupled GPCR structures (***Kim et al., 2020***; ***Yuan et al., 2021***; ***Zhuang et al., 2022***; ***Hua et al., 2020***; ***Rasmussen et al., 2011***; ***Kato et al., 2019***; ***Figure 3B***). Nevertheless, the Gαi structure adopts a nucleic acid-free state, similar to the μOR-Gi complex (***Figure 3—figure supplement 1***; ***Zhuang et al., 2022***; ***Wall et al., 1995***). The interacting surface area between the receptor and Gαi

subunit is 1,196 Å$^2$, which is comparable to those in other GPCR-G$_i$ complexes (μOR : 1,260 Å$^2$. NTS$_1$ : 1,197 Å$^2$. and S1P$_1$: 1,376 Å$^2$. *Yuan et al., 2021*; *Zhuang et al., 2022*; *Kato et al., 2019*). As described above, ICL2 and TM7 extensively interact with the αN and α5 helix of the Gα$_i$ subunit, respectively. These interactions are uniquely observed in the ET$_B$-G$_i$ complex and can compensate for the shallow binding of G$_i$.

## Discussion

In this study, we established the Fusion-G system, which facilitates the efficient expression and purification of the GPCR-G-protein complex. Additionally, the ability to monitor the fluorescence of GFP fused in the C-terminus of the receptor makes it easier to optimize the expression and purification conditions. The tethering strategy only increases the proportion of the complex by constantly placing the G-protein around the receptor rather than tightly anchoring the complex, suggesting that the fusion-G-system has minimal effect on the complex structure. It should be noted that the potential artifacts resulting from the tethering strategy cannot be entirely ruled out. The hybrid approach we have developed, combining NanoBiT tethering and FSEC methodology, is an effective option for comprehensive structural analysis of various other membrane protein complexes.

The ET$_B$-G$_i$ structure determined by this strategy filled in the last piece of the puzzle, and our understanding of the mechanism of receptor activation has been significantly deepened. The ligand binding pocket is essentially similar in both the crystal structure of the receptor alone and in the G$_i$ protein complex (*Figure 2A*). However, the W336$^{6.48}$ rotamer, which comprises the bottom of the ligand-binding pocket, differs (*Figure 2B*), thereby leading to the activation of the intracellular side of the receptor (*Figure 2C and D*). Thus, it may be crucial to choose and evaluate compounds based on the bottom of the ligand pocket in the current ET$_B$-G$_i$ structure (*Figure 2B*) to develop small molecule ET$_B$ agonists for neuroprotection and cancer therapy.

Previous structural analyses have demonstrated that docking the α5-helix to the receptor cavity leads to destabilization of the nucleic acid binding site located at the root of the α5-helix (*Figure 3— figure supplement 1*; *Zhuang et al., 2022*; *Rasmussen et al., 2011*; *Su et al., 2020*), thus promoting the GDP/GTP exchange reaction. While the binding of Gα$_s$ remains mostly indistinguishable in all complexes, the binding of Gα$_i$ varies, displaying different Gα rotations in relation to the receptor *Okamoto et al., 2021*. Nevertheless, the depth of the docking towards the receptor remains constant (*Figure 3B*). Since the N$^{7.49}$P$^{7.50}$xxY$^{7.53}$ motif is not conserved in ET$_B$, TM7 undergoes the downward movement upon the G$_i$ binding, resulting in the shallow position of the α5-helix. Thus, the restrictions of the residues in the α5-helix may be less strict, accounting for the G-protein promiscuity of ET$_B$. Since the other promiscuous GPCRs (e.g. NTS$_1$) preserve the N$^{7.49}$P$^{7.50}$xxY$^{7.53}$ motif, the proposed mechanism for G-protein promiscuity may be unique to ET$_B$. Such information is essential for understanding the mechanism of G-protein activation by GPCRs.

## Materials and methods

**Key resources table**

| Reagent type (species) or resource | Designation | Source or reference | Identifiers | Additional information |
|---|---|---|---|---|
| peptide, recombinant protein | ET-1 | PEPTIDE INSTITUTE, INC. | Cat #, 4198 v | Ligand for ET$_B$ |
| Other | Sf-900 II SFM | Thermo Fisher Scientific | Cat #, 10902088 | Expression medium for sf9 cells |
| chemical compound, drug | n-dodecyl-β-D-maltoside | Calbiochem | CAS number: 69227-93-6 | Detergents used in purification of ET$_B$-G$_i$ complex |
| chemical compound, drug | Cholesteryl hemisuccinate | Merck Millipore | CAS number: 1510-21-0 | For purifying ET$_B$-G$_i$ complex |
| peptide, recombinant protein | Apyrase | New England Biolabs | Cat #, M0398 | Enzyme used for ET$_B$-G$_i$ complex formation |

*Continued on next page*

*Continued*

| Reagent type (species) or resource | Designation | Source or reference | Identifiers | Additional information |
|---|---|---|---|---|
| Other | Anti-DYKDDDDK G1 Affinity resin | Gen Script | Cat #, L00432 | Affinity resin for DYKDDDDK tags |
| chemical compound, drug | Lauryl Maltose Neopentyl Glycol | Anatrace | CAS number: 1257852-96-2 | Detergents used in purification of $ET_B$-$G_i$ complex |
| Software, algorithm | EPU | Thermo Fisher Scientific | | For Cryo-EM data collection |
| Software, algorithm | RELION-3.1 | *Zivanov et al., 2018* | RRID:SCR_016274 | For Cryo-EM data processing |
| Software, algorithm | cryoSPARC v3.3 | STRUCTURA BIOTECHNOLOGY | RRID:SCR_016501 | For cryo-EM data processing |
| Software, algorithm | Coot | *Emsley et al., 2010* | RRID:SCR_014222 | For structure model building |
| Software, algorithm | Phenix 1.19–4092 | *Afonine et al., 2018* | RRID:SCR_014224 | For structure refinement |
| other | Quantifoil holey carbon grid | Quantifoil | R1.2/1.3, Au, 300 mesh | For cryo-EM specimen preparation |

## Constructs

The full-length human $ET_B$ gene was subcloned into the pFastBac vector with an HA-signal peptide sequence on its N-terminus and the LgBiT fused to its C-terminus followed by a 3 C protease site and EGFP-His8 tag. A 15 amino sequence of GGSGGGGSGGSSSGG was inserted into both the N-terminal and C-terminal sides of LgBiT. The Flag epitope tag (DYKDDDDK) was introduced between G57 and L66. The native signal peptide was replaced with the haemagglutinin signal peptide. Rat $G\beta_1$ and bovine $G\gamma_2$ were subcloned into the pFastBac Dual vector, as described previously *Kobayashi et al., 2020*. In detail, rat $G\beta_1$ was cloned with a C-terminal HiBiT connected with a 15 amino sequence of GGSGGGGSGGSSSGG. Moreover, human $G\alpha_{i1}$ was subcloned into the C-terminus of the bovine $G\gamma_2$ with a nine amino sequence GSAGSAGSA linker. The resulting pFastBac dual vector can express the $G_i$ trimer.

## Complex formation and FSEC analysis

Bacmid preparation and virus production was performed according to the Bac-to-Bac baculovirus system manual (Gibco, Invitrogen). *Spodoptera frugiperda* Sf9 cells at a density of $3 \times 10^6$ cells/ml were co-infected with baculoviruses encoding receptor and $G_i$ trimer at the ratio of 1:1. For the expression of the receptor alone, the baculovirus encoding receptor was only used. Cells were harvested 48 hr after infection. 1 ml cell pellets were solubilized in 200 µl buffer, containing 20 mM Tris-HCl, pH 8.0, 150 mM NaCl, 1% n-dodecyl-β-D-maltoside (DDM) (Calbiochem), 0.2% cholesteryl hemisuccinate (CHS) (Merck) and rotated for 1 hr at 4 °C.

For the complex formation with the agonist, cell pellets were resuspended in 20 mM Tris-HCl, pH 8.0, 100 mM NaCl, and 10% Glycerol, and homogenized by douncing ~20 times. Apyrase was added to the lysis at a final concentration of 25 mU/ml. Each agonist was added at a final concentration of 10 µM. The homogenate was incubated at room temperature for 1 hr with flipping. Then, DDM and CHS were added to a final concentration of 1% and 0.2%, respectively for 1 hr at 4 °C.

The supernatants were separated from the insoluble material by ultracentrifugation at 100,000 g for 20 min. A fraction of the resulting supernatant (10 µl) was loaded onto a Superdex 200 increase 10/300 column in 20 mM Tris-HCl, pH 8.0, 150 mM NaCl, and 0.03% DDM, and run at the flow rate of 0.5 ml/min. The eluent was detected by a fluorometer with the excitation wavelength (480 nm) and emission wavelength settings (512 nm).

## ET-1–$ET_B$–$G_i$ complex formation and purification

For expression, 300 ml of the Sf9 cells at a density of $3 \times 10^6$ cells/ml were co-infected with baculovirus encoding the $ET_B$-LgBiT-EGFP and $G_i$ trimer at the ratio of 1:1. Cells were harvested 48 hr after

infection. Cell pellets were resuspended in 20 mM Tris-HCl, pH 8.0, 100 mM NaCl, and 10% Glycerol, and homogenized by douncing ~30 times. Apyrase was added to the lysis at a final concentration of 25 mU/ml. ET-1 was added at a final concentration of 2 μM. The lysate was incubated at room temperature for 1 hr with flipping. Then, the membrane fraction was collected by ultracentrifugation at 180,000 g for 1 hr. The cell membrane was solubilized in buffer, containing 20 mM Tris-HCl, pH 8.0, 150 mM NaCl, 1% DDM, 0.2% CHS, 10% glycerol, and 2 μM ET-1 for 1 hr at 4 °C. The supernatant was separated from the insoluble material by ultracentrifugation at 180,000 g for 30 min and then incubated with the Anti-DYKDDDDK G1 resin (Genscript) for 1 hr. The resin was washed with 20 column volumes of wash buffer containing 20 mM Tris-HCl, pH 8.0, 500 mM NaCl, 10% Glycerol, 0.1% Lauryl Maltose Neopentyl Glycol (LMNG) (Anatrace), and 0.01% CHS. The complex was eluted by the wash buffer containing 0.15 mg ml$^{-1}$ Flag peptide. The eluate was treated with 0.5 mg of HRV-3C protease (homemade) and dialyzed against buffer (20 mM Tris-HCl, pH 8.0, and 300 mM NaCl). Then, cleavaged GFP-His$_8$ and HRV-3C protease were removed with Ni$^+$-NTA resin. The flow-through was incubated with the scFv16, prepared as described previously *Okamoto et al., 2021*. The complex was concentrated and loaded onto a Superdex 200 increase 10/300 column in 20 mM Tris-HCl, pH 8.0, 150 mM NaCl, 0.01% LMNG, 0.001% CHS, and 1 μM agonist. Peak fractions were concentrated to 8 mg/ml.

### Cryo-EM grid preparation and data acquisition

The purified complex was applied onto a freshly glow-discharged Quantifoil holey carbon grid (R1.2/1.3, Au, 300 mesh) and plunge-frozen in liquid ethane by using a Vitrobot Mark IV (FEI). Cryo-EM data collection was performed on a 300 kV Titan Krios G3i microscope (Thermo Fisher Scientific) equipped with a BioQuantum K3 imaging filter (Gatan) and a K3 direct electron detector (Gatan). In total, 10,408 movies were acquired with a calibrated pixel size of 0.83 Å pix$-1$ and with a defocus range of −0.8 to −1.6 μm, using the EPU software (Thermo Fisher's single-particle data collection software). Each movie was acquired for 2.3 s and split into 48 frames, resulting in an accumulated exposure of about 49.965 e− Å$-2$.

All acquired movies in super-resolution mode were binned by 2x and were dose-fractionated and subjected to beam-induced motion correction implemented in RELION 3.1 *Zivanov et al., 2018*. The contrast transfer function (CTF) parameters were estimated using patch CTF estimation in cryoSPARC v3.3 *Punjani et al., 2017*. Particles were initially picked from a small fraction with the Blob picker and subjected to several rounds of two-dimensional (2D) classification in cryoSPARC. Selected particles were used for training of topaz model *Bepler et al., 2019*. For the full dataset, 3,863,134 particles were picked and extracted with a pixel size of 3.32 Å, followed by 2D classification to remove carbon edges and ice contaminations. A total of 1,442,243 particles were re-extracted with the pixel size of 1.16 Å and curated by three-dimensional (3D) classification without alignment in RELION. Finally, the 260,085 particles in the best class were reconstructed using non-uniform refinement, resulting in a 2.80 Å overall resolution reconstruction, with the gold standard Fourier Shell Correlation (FSC = 0.143) criteria in cryoSPARC. Moreover, the 3D model was refined with a mask on the receptor. As a result, the local resolution of the receptor portion improved with a nominal resolution of 3.13 Å. The local resolution was estimated by cryoSPARC. The processing strategy is described in *Figure 1— figure supplement 2*.

### Model building and refinement

The quality of the density map was sufficient to build an atomic model. Previously reported high-resolution crystal structure of the ET-3 bound ET$_B$ receptor (PDB 6IGK) and cryo-EM MT$_1$-G$_i$ structure (PDB 7DB6) were used as the initial models for the model building of receptor and G$_i$ portions, respectively *Shihoya et al., 2018*; *Okamoto et al., 2021*. Initially, the models were fitted into the density map by jiggle fit using COOT *Emsley et al., 2010*. Then, atomic models were readjusted into the density map using COOT and refined using phenix.real_space_refine (v1.19) with the secondary structure restraints using phenix.secondary_structure_restraints *Adams et al., 2010*; *Afonine et al., 2018*.

### Acknowledgements

We thank K Ogomori and C Harada for their technical assistance. This work was supported by JSPS KAKENHI grants 21H05037 (ON), 22K19371 and 22H02751 (WS); ONO Medical Research Foundation

(WS); The Kao Foundation for Arts and Sciences (WS); The Takeda Science Foundation (WS); The Uehara Memorial Foundation (WS); the Platform Project for Supporting Drug Discovery and Life Science Research (Basis for Supporting Innovative Drug Discovery and Life Science Research (BINDS)) from AMED, under grant numbers JP22ama121012 and JP22ama121002 (support number 3272).

## Additional information

### Competing interests

Osamu Nureki: is a co-founder and scientific advisor for Curreio. The other authors declare that no competing interests exist.

### Funding

| Funder | Grant reference number | Author |
|---|---|---|
| Japan Society for the Promotion of Science | 21H05037 | Osamu Nureki |
| Japan Society for the Promotion of Science | 22K19371 | Wataru Shihoya |
| Japan Society for the Promotion of Science | 22H02751 | Wataru Shihoya |
| Ono Medical Research Foundation | | Wataru Shihoya |
| Kao Foundation for Arts and Sciences | | Wataru Shihoya |
| Takeda Science Foundation | | Wataru Shihoya |
| Uehara Memorial Foundation | | Wataru Shihoya |
| Platform Project for Supporting Drug Discovery and Life Science Research | JP22ama121012 (support number 3272) | Wataru Shihoya |
| Platform Project for Supporting Drug Discovery and Life Science Research | JP22ama121002 (support number 3272) | Wataru Shihoya |

The funders had no role in study design, data collection and interpretation, or the decision to submit the work for publication.

### Author contributions

Fumiya K Sano, Data curation, Software, Formal analysis, Validation, Visualization, Writing – original draft; Hiroaki Akasaka, Data curation, Validation, Methodology, Writing – review and editing; Wataru Shihoya, Conceptualization, Resources, Supervision, Funding acquisition, Validation, Methodology, Writing – original draft, Project administration; Osamu Nureki, Conceptualization, Supervision, Funding acquisition, Project administration, Writing – review and editing

### Author ORCIDs

Fumiya K Sano ⓘ http://orcid.org/0000-0002-8965-788X
Hiroaki Akasaka ⓘ http://orcid.org/0000-0003-2118-0912
Wataru Shihoya ⓘ http://orcid.org/0000-0003-4813-5740
Osamu Nureki ⓘ http://orcid.org/0000-0003-1813-7008

Decision letter and Author response
Decision letter https://doi.org/10.7554/eLife.85821.sa1
Author response https://doi.org/10.7554/eLife.85821.sa2

## Additional files

### Supplementary files
• Transparent reporting form

### Data availability
Cryo-EM Density maps and structure coordinates have been deposited in the Electron Microscopy Data Bank (EMDB) and the Protein Data Bank (PDB), with accession codes EMD-35814 and PDB 8IY5 for the $ET_B$-$G_i$ complex, and EMD-35815 and PDB 8IY6 for the $ET_B$-$G_i$ complex (Receptor focused).

The following datasets were generated:

| Author(s) | Year | Dataset title | Dataset URL | Database and Identifier |
|---|---|---|---|---|
| Sano FK, Akasaka H, Shihoya W, Nureki O | 2023 | ETB-Gi complex bound to Endotheline-1, focused on receptor | https://www.ebi.ac.uk/emdb/EMD-35815 | Electron Microscopy Data Bank, EMD-35815 |
| Sano FK, Akasaka H, Shihoya W, Nureki O | 2023 | ETB-Gi complex bound to endothelin-1 | https://www.ebi.ac.uk/emdb/EMD-35814 | Electron Microscopy Data Bank, EMD-35814 |
| Sano FK, Akasaka H, Shihoya W, Nureki O | 2023 | ETB-Gi complex bound to Endotheline-1, focused on receptor | https://www.rcsb.org/structure/8IY6 | RCSB Protein Data Bank, 8IY6 |
| Sano FK, Akasaka H, Shihoya W, Nureki O | 2023 | ETB-Gi complex bound to endothelin-1 | https://www.rcsb.org/structure/8IY5 | RCSB Protein Data Bank, 8IY5 |

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
