## [Editor Report]

By combining NanoBiT tethering and FSEC methodology, the authors present a useful strategy for the efficient expression and purification of the GPCR-G-protein complex, as demonstrated here by the cryo-EM structure of human ETB in complex with the vasoconstrictioning peptide ET-1 and the inhibitory G-protein (Gi). The ETB-Gi-protein complex provides valuable and convincing data on how ET-1 binding is coupled to Gi-protein binding. The complex structure is solid and will appeal to the GPCR and pharmacology communities.

---

## [Decision Letter]

**Decision letter after peer review:**

Thank you for submitting your article "Cryo-EM structure of the endothelin-1-ETB-Gi complex" for consideration by *eLife*. Your article has been reviewed by 3 peer reviewers, including David Drew as Reviewing Editor and Reviewer #1, and the evaluation has been overseen by Kenton Swartz as the Senior Editor. The following individual involved in review of your submission has agreed to reveal their identity: Arun Shukla (Reviewer #3).

Essential revisions:

Overall, this is an excellent manuscript. Please consider the following comments in a revised version of the manuscript.

1. Regarding Figure 3B. It might true helpful to also include the amount of buried SA of the Gi interaction to better compare the extent of interaction between a5 and the different solved G-protein complex structures.

2. Authors highlight a shallow binding modality of alpha5 helix on the receptor and also corelate it with previous studies. It would be helpful if they also comment the functional implications, if any, for this observation? For example, does it have a role in lose G-protein subtype selectivity of ETbR compared to other GPCR where a deeper insertion is observed?

3. Please consider commentsing on the use of NanoBIT and if this coupling could influence lipid binding, such as PIP2? Also, we would invite the discussion of the use of this technology in the manuscript for other membrane protein complexes.

4. Please consider expanding the discussion of their findings beyond the structural biology point of view and the potential implications of this work from a clinical point of view in the background and discussion to add more depth to this manuscript.

5. "Junk particles" are mentioned in the method section on cryo-EM data processing. While we all use this term in our conversations, I wonder if a better terminology can be used when describing such particles in a research manuscript.

*Reviewer #2 (Recommendations for the authors):*

The study by Sano et al. observed detailed G-protein activation by endothelin receptor type B after its activation. This is an important study of an underreported topic that revealed novel findings that could be potentially valuable in various disciplines. However, the reviewer feels that the authors would be better served to expand on the discussion of their findings beyond the structural biology point of view. It would be better if the authors put clinical point of view in the background and discussion to add more value in this manuscript.

---

## [Author Response]

Essential revisions:Overall, this is an excellent manuscript. Please consider the following comments in a revised version of the manuscript.1. Regarding Figure 3B. It might true helpful to also include the amount of buried SA of the Gi interaction to better compare the extent of interaction between a5 and the different solved G-protein complex structures.

To correctly calculate the buried surface area, we modeled the ICL2 and ICL3. It should be noted that the density corresponding to ICL3 is weak, not enough to model the side chains and discuss the specific ICL3-G_i_ interaction. Thus, we corrected the description of ICL2 as:

“Moreover, ICL2 forms extensive van der Waals interactions with the αN (Figure S3c), which are not observed in other GPCR-G_i_ complexes” (lines 183 to 185), and added the discussion about the buried surface area as:

“The interacting surface area between the receptor and Gα_i_ subunit is 1,196 Å2, which is comparable to those in other GPCR-G_i_ complexes (µOR:1,260 Å2. NTS_1_:1,197 Å2. and S1P_1_: 1,376 Å2.^34,38,49^). As described above, ICL2 and TM7 extensively interact with the αN and α5 helix of the Gα_i_ subunit, respectively. These interactions are uniquely observed in the ET_B_-G_i_ complex and can compensate for the shallow binding of G_i_.” (lines 193 to 198).

2. Authors highlight a shallow binding modality of alpha5 helix on the receptor and also corelate it with previous studies. It would be helpful if they also comment the functional implications, if any, for this observation? For example, does it have a role in lose G-protein subtype selectivity of ETbR compared to other GPCR where a deeper insertion is observed?

We added the functional implication about the shallow Gi coupling as:

“Since the N7.49P7.50xxY7.53 motif is not conserved in ETB, TM7 undergoes the downward movement upon the Gi binding, resulting in the shallow position of the α5-helix. Thus, the restrictions of the residues in the α5-helix may be less strict, accounting for the G-protein promiscuity of ETB. Since the other promiscuous GPCRs (e.g., NTS1) preserve the N7.49P7.50xxY7.53 motif, the proposed mechanism for G-protein promiscuity may be unique to ETB” (lines 226 to 233).

3. Please consider commentsing on the use of NanoBIT and if this coupling could influence lipid binding, such as PIP2? Also, we would invite the discussion of the use of this technology in the manuscript for other membrane protein complexes.

In this study, linkers of sufficient length are inserted between LgBiT and the receptor and between HiBiT and Gβ, respectively. We suppose tethering strategy does not tightly anchor the complex, but only contributes to increasing the proportion of the complex by locate G protein around the receptor constantly. Recently, several papers have employed NanoBiT tethering strategies. In most of them, no density corresponding to NanoBiT has been observed, however, we suppose the tethering strategy also would have had little effect on the structure of the complex in this study. We added the above discussion as:

“The tethering strategy only increases the proportion of the complex by constantly placing the G protein around the receptor rather than tightly anchoring the complex, suggesting that the fused G system has minimal effect on the complex structure. It should be noted that the potential artifacts resulting from the tethering strategy cannot be entirely ruled out. The hybrid approach we have developed, combining NanoBiT tethering and FSEC methodology, is an effective option for comprehensive structural analysis of various other membrane protein complexes.” (lines 204 to 211).

4. Please consider expanding the discussion of their findings beyond the structural biology point of view and the potential implications of this work from a clinical point of view in the background and discussion to add more depth to this manuscript.

We appreciate reviewer’s constructive remarks. Based on the suggestion, we added clinical details of endothelins in the introduction (lines 77 to 92), and the relationship between our study and clinical application in discussion (lines 212 to 220).

5. "Junk particles" are mentioned in the method section on cryo-EM data processing. While we all use this term in our conversations, I wonder if a better terminology can be used when describing such particles in a research manuscript.

Based on the suggestion, we corrected the wording to carbon edge and ice contamination (line 310).

Reviewer #2 (Recommendations for the authors):The study by Sano et al. observed detailed G-protein activation by endothelin receptor type B after its activation. This is an important study of an underreported topic that revealed novel findings that could be potentially valuable in various disciplines. However, the reviewer feels that the authors would be better served to expand on the discussion of their findings beyond the structural biology point of view. It would be better if the authors put clinical point of view in the background and discussion to add more value in this manuscript.

We appreciate reviewer’s constructive remarks. Based on the suggestion, we added clinical details of endothelins in the introduction (lines 77 to 92), and the relationship between our study and clinical application in discussion (lines 212 to 220).